# A Comparative Analysis of Pain Assessment Methods in the Initial Postoperative Phase Following Different Pilonidal Cyst Surgeries

**DOI:** 10.3390/medicina60101710

**Published:** 2024-10-18

**Authors:** Edvinas Dainius, Julija Garnyte, Egle Juskeviciute, Audrius Parseliunas, Tadas Latkauskas, Guoda Burzinskiene, Donatas Venskutonis, Algimantas Tamelis

**Affiliations:** 1Department of Surgery, Lithuanian University of Health Sciences, LT-50161 Kaunas, Lithuania; 2Faculty of Medicine, Lithuanian University of Health Sciences, LT-50161 Kaunas, Lithuania; 3Department of Obstetrics and Gynaecology, Lithuanian University of Health Sciences, LT-50161 Kaunas, Lithuania

**Keywords:** postoperative pain, pain assessment, visual analogue scale, pressure algometry, pilonidal disease

## Abstract

*Background and Objectives:* In this study, we aimed to evaluate pain intensity in patients after pilonidal disease surgeries of varying extent using pressure algometry and the visual analog scale and to explore potential correlations between these methods. *Materials and Methods:* A total of 78 adult patients with symptomatic pilonidal cysts were enrolled in this study. The patients were divided into two groups based on the type of surgery assigned to each patient at the pre-hospital consultation: pit-picking surgery (*n* = 39) and radical excision (*n* = 39). The pain levels at the surgical site were assessed and compared using the visual analog scale (VAS) and pressure algometry the morning before surgery and the day after the operation. *Results:* There was no statistically significant difference (*p* > 0.05) in VAS measurement results between surgical groups when comparing pain intensity experienced by patients before, during, and after surgery. Notably, specific pressure algometry variables (pressure pain tolerance left 2.05 ± 1.46 compared to 1.42 ± 0.73 kg/cm^2^, *p* = 0.02; maximum pressure pain tolerance left 2.91 ± 1.33 compared to 2.32 ± 1.14 kg/cm^2^, *p* = 0.04; maximum pressure pain tolerance center 2.51 ± 1.07 compared to 1.91 ± 0.91 kg/cm^2^, *p* = 0.01; interval of pressure pain tolerance center 0.98 ± 0.62 compared to 0.59 ± 0.39 kg/cm^2^, *p* = 0.00) on the first postoperative day were significantly lower in the “pit-picking” group compared to the excision group. Furthermore, no statistically significant correlation was found between VAS and pressure algometry measurements either before surgery or on the first postoperative day. *Conclusions:* In the early postoperative period following pilonidal disease surgery of varying extents, pain measured with the VAS does not differ. In contrast, the pressure algometry method showed greater pain in the minimally invasive surgery cohort on the first postoperative day. However, further larger studies are needed to compare these pain assessment methods in reporting pain intensity experienced during patient movement.

## 1. Introduction

Pain, as defined by the International Association for the Study of Pain (ISAP), is a complex and unpleasant sensory and emotional experience associated with potential or actual tissue damage [1]. It is noteworthy that accurately and objectively assessing pain intensity, which can vary subjectively among patients, remains a considerable challenge. The inability of some patients to verbally express their pain levels prompted scientists to come up with non-verbal ways of pain assessment [1]. Consequently, various methods of pain assessment have been introduced to medical communities over the years. 

The visual analog scale (VAS) is a simple and widely used method for pain assessment. It consists of a 100 mm line, with 0 representing no pain and 100 representing the “worst imaginable pain”. This method is cost-effective and convenient, allowing healthcare professionals to monitor pain reduction, compare pain levels between patients with similar conditions, and provide a quick way for patients to report their pain [2,3]. However, the VAS also presents challenges in its objective application. Patients must have sufficient cognitive abilities to comprehend the concept of the scale and the instructions provided by the healthcare professional, as well as the physical ability to mark their response on paper. Additionally, the interpretation of VAS readings has limitations, as pain is subjective and identical scores may be interpreted differently based on individual patients, the etiology of their pain, and its duration [3,4,5].

Given the limitations of the VAS, alternative non-invasive methods have been created to objectify the pain intensity reporting experience. Pressure algometry, for instance, involves applying pressure to specific pain points to measure the pressure pain threshold (PPT). Pressure algometry enables objective assessment of the pain levels experienced by patients at specific anatomical sites. It has been proven to be an effective method in diagnosing temporomandibular or knee dysfunctions [6,7]. However, this pain assessment method is rarely used in clinical practice. Specific conditions, such as the invariable position of the algometer, the dynamics of the exerted pressure, and the exact location to which pressure is applied, have to be maintained for each measurement [7]. Nevertheless, attempts have been made to combine pressure algometry and other pain assessment scales to make pain evaluation less subjective [8].

Effective postoperative pain management is progressively recognized as one of the key factors in the successful treatment of surgical patients [4,9]. Poor postsurgical pain control can lead to delayed healing, a longer hospital stay, a later return to work and everyday activities, patient dissatisfaction, and psychological challenges [10,11]. A detailed assessment of preoperative pain can impact the surgical approach and scope of intervention, as well as lower the likelihood of postsurgical complications and chronic pain [11,12].

One specific surgical area is the sacrococcygeal region. Various surgical techniques have been developed for managing pilonidal disease (PD). Radical excision is the most common surgical treatment for this disease [13,14]. Postsurgical pain may arise, particularly during the first few days after surgery, especially during wound dressing changes that are essential for preventing infections [15]. To catalyze patients’ recovery and reduce the risk of complications, minimally invasive surgical methods have been introduced to treat primary PD. During “pit-picking” surgery, the wound is sutured, which may cause additional pain during the first few days of recovery as tension is created in the tissues [16]. Although these procedures are routinely performed, they are associated with many complications, one of which is pelvic and perineal pain during specific movements such as sitting or defecation [17]. This pain is unavoidable with early mobilization after surgery, which is a critical component of fast and successful recovery [18]. 

Therefore, it is essential to identify the potency of pain early after surgery. The present study aimed to evaluate pain intensity in patients after pilonidal disease surgeries of varying extent using pressure algometry and the visual analog scale and to explore potential correlations between these methods. The secondary objective of this study was to assess the need for analgesics during the first 24 h following minimally invasive or excision surgery.

## 2. Materials and Methods

### 2.1. Study Setting

This study is a prospective comparative trial conducted at the Department of Surgery of the Lithuanian University of Health Sciences (LUHS) Kaunas Hospital. Following approval from the local ethics committee (Approval No. BE-2-61, 18 August 2020), patients referred to the Department of Surgery from 1 September 2020 to 1 February 2023 for elective pilonidal disease surgery were invited to participate after giving written informed consent.

### 2.2. Participants

A total of 78 patients were included in the study based on the following eligibility criteria: age above 18 years, diagnosis of chronic symptomatic pilonidal cyst, voluntary consent to participate, signed informed consent form, and the ability to communicate verbally. The exclusion criteria were as follows: presented with an acute or asymptomatic cyst, non-Lithuanian speakers, diagnosed with cognitive, visual, auditory, or locomotor system disorders, and refusal to participate in the study.

After admission to the hospital, patients were informed about the study’s objectives and organizational aspects. The patients were divided into two study groups based on the type of surgery assigned to each patient at the pre-hospital consultation: pilonidal cyst excision surgery (N = 39) or minimally invasive “pit-picking” surgery (N = 39). All patients underwent surgery with spinal anesthesia.

### 2.3. Sample Size

When calculating the sample volume, we considered postoperative pain after 1 day as the primary endpoint. Past studies suggest an average postoperative pain score of 2.5 ± 1.7 SD using the VAS after radical surgery without suturing [19]. We will consider a clinically significant difference between the groups when the pain in the minimally invasive surgery group is 50% less 1 day after the procedure. To calculate the sample size of the study, the probability of the first type of error was assumed to be α = 0.01 and β = 0.2 for the second type, and the statistical power of the study = 0.8. Therefore, 39 patients per group were required.

### 2.4. Surgical Procedures

During minimally invasive pit-picking surgery, the midline pits and secondary fistula openings were examined to determine the sinus tracts’ direction and length. A 3 or 5 mm punch biopsy needle was used to remove the pits. The pores and underlying tissue with hair follicles were excised. A 2 cm lateral incision was made near the pilonidal cyst to drain the cyst cavity. Nonabsorbable sutures were used to close the wounds. The lateral incision was left unsutured.

During radical excision surgery, the pilonidal cyst cavity, along with the sinus tracts and fistulas, was outlined using brilliant green dye. A symmetrical elliptical incision was made to remove the pits and fistula openings. Monopolar electrocautery was used to excise the pilonidal cyst from the surrounding healthy tissue and sacrococcygeal fascia. The wound was left open for secondary intension healing.

### 2.5. Study Design

For all patients in the study, the following variables were assessed and measured: sex, age, body mass index, smoking status, the duration of symptoms, ASA class, primary PD diagnosis, the size of PD, the count of grown follicles, and the presence of fistulas.

All patients completed a visual analog scale on the morning of admission before the operation. At 9 a.m., while resting in the ward, patients were instructed to indicate their current pain level on a 100 mm line VAS, ranging from “no pain” to “unbearable pain”. The distance from “no pain” to the point marked by the patient was measured using a ruler and converted into a numerical value to the nearest millimeter, indicating the intensity of pain felt by the patient. The same assessment was repeated the next day after surgery at 9 a.m. During the VAS assessment, patients were additionally asked to identify the pain intensity they felt during the injection of spinal anesthesia.

Simultaneously, pressure algometry measurements were performed using a handheld pressure algometer (Wagner “Pain test FPX”, Greenwich). Pressure was applied perpendicular to the body’s surface at a rate of 0.5 kgf/s. Pressure pain threshold (PPT) was determined when the sensation of pressure turned into a feeling of pain. Meanwhile, the maximally tolerated pressure pain (MTPP) was noted when the tolerable pain reached an unbearable level. The interval of pressure pain tolerance (IPPT) was calculated as the difference between the MTPP and PPT. Preoperatively, pressure-induced pain was assessed on both sides, 2 cm to the left and 2 cm to the right of the pilonidal cyst, and in the middle of the pilonidal cyst in the sacrum. Postoperatively, pressure pain was measured on both sides of the surgical wound and in the middle of the wound through a sterile dressing, as demonstrated in Figure 1. Both patient groups underwent the same pain assessment procedure, and the results were compared to identify statistically significant differences.

### 2.6. Pain Management

On the day of surgery, once the spinal analgesia wore off, participants reporting a pain score of 30 or higher on the VAS received intramuscular (IM) injections of 100 mg/2 mL ketoprofen.

Additional IM doses of 100 mg/2 mL ketoprofen were administered to those with a pain score of 30 or higher during their first postoperative day. For continued pain management at home, patients were prescribed dexketoprofen 25 mg if the pain persisted or worsened.

### 2.7. Statistical Analysis

Data analysis was conducted using Statistical Package for the Social Sciences (SPSS, version 29.0, IBM Corp., Armonk, NY, USA). Categorical variables were described by presenting the frequency of their values as a percentage. Differences in values between groups were assessed using the Pearson chi-squared test for categorical data. The Shapiro–Wilk test was used to determine if the quantitative data were distributed normally with respect to the mean. Normally distributed variables were compared using Student’s *t*-test for independent variables. Standard deviations (SDs) and mean values were used to describe the data. Non-parametric data were compared using the Mann–Whitney U test for independent variables. Non-parametric data were expressed as median values with 95% confidence intervals. Spearman’s correlation coefficient was used to determine the relationship between the two pain assessment methods. A confidence level of 95% (*p* < 0.05) was chosen to draw conclusions.

## 3. Results

### 3.1. Baseline Characteristics of Patients and Peculiarities of Pilonidal Disease

The process of enrollment is depicted in Figure 2. Baseline characteristics were compared between the two patient groups, and no statistically significant differences were observed (Table 1).

### 3.2. Pain Intensity Levels Measured by Visual Analogue Scale

While analyzing the pain intensity measurements using the VAS method before and after surgery, it was determined that the variables were unevenly distributed. Therefore, the results are presented as medians with 95% confidence intervals.

No statistically significant difference (*p* > 0.05) between surgical groups was determined in the VAS measurement results when comparing the pain intensity experienced by the patients before surgery, after surgery, and during anesthesia (Table 2).

Regarding the need for analgesics, no statistically significant difference was found between the two patient groups in terms of the mean doses of analgesics consumed on surgery day (excision group 1.2 ± 0.16 compared to “pit-picking” group 1.38 ± 0.17 (mean ± SD), *p* = 0.43) and on the first postoperative day (excision group 0.79 ± 0.15 compared to “pit-picking” group 0.61 ± 0.15 (mean ± SD), *p* = 0.38).

### 3.3. Pain Intensity Levels Measured by Pressure Algometry

Pressure algometry was used to measure the pain intensity experienced by patients before and after surgery (Table 3 and Table 4). A statistically significant difference (*p* < 0.05) in pain intensity levels was determined between the surgical groups one day after surgery when comparing the pressure pain tolerance (PPT) on the left, the maximum pressure pain tolerance (MTPP) on the left and at the center, and the interval of pressure pain tolerance (IPPT) at the center of the surgical wound. These variables were significantly lower in the “pit-picking” group than in the excision group, which means that in the minimally invasive group, pain after surgery was greater and tolerated worse. (Table 4).

### 3.4. Correlation between VAS and Pressure Algometry

This study aimed to compare the VAS and pressure algometry to determine the potential correlation between these two different measurements of pain intensity at various pain points before and after surgery. By applying Spearman’s correlation coefficient, no statistically significant correlation (*p* > 0.05) was found between VAS and pressure algometry measurements for PPT, MTPP, and PPTI before and on the first day after surgery at different pain points.

In summary, no statistically significant correlation was found between VAS and pressure algometry measurements either before surgery or on the first postoperative day.

## 4. Discussion

The goal of this study was to compare pressure algometry and the visual analog scale in measuring pain intensity in patients after PD surgery. This study included two homogeneous groups of patients diagnosed with pilonidal disease who were referred to our hospital for surgical treatment and underwent either minimally invasive “pit-picking” surgery or radical excision.

Assessment of pain intensity remains a vital component of both pre- and postsurgical management of pilonidal disease, as it significantly impacts patients’ quality of life and functional capabilities [4,9,11]. By effectively managing perioperative pain, healthcare providers can improve patients’ well-being, facilitating their recovery and return to daily activities and reducing the physiological and psychological burden associated with chronic pain, delayed wound healing, and prolonged hospitalization [10,11]. Preoperative pain assessment allows for the tailoring of anesthetic and analgesic strategies to individual needs, minimizing perioperative discomfort, while postoperative pain control is crucial for ensuring patient recovery and satisfaction and addressing any complications early [11,12]. Ensuring accurate assessment and effective management of pain is essential for achieving the best possible outcomes for patients undergoing treatment for pilonidal disease [12].

Two different surgical methods of varying extents were used to treat pilonidal disease in this study. Previous studies have shown that less invasive surgical treatment for PD has advantages in terms of reduced postoperative pain [20,21]. However, in our study, postsurgical pain evaluated by the VAS did not differ between the two groups (*p* > 0.05). This may be because pain assessment using the VAS was conducted at rest on the first day after the operation, while postsurgical patients after pilonidal cyst surgery usually experience pain when sitting or using the bathroom [12]. Additionally, some studies show that pain levels in the minimally invasive group are significantly lower during the later postoperative phase, particularly one week after surgery [22]. After radical excision surgery, the wound is not tense, and patients report minimal pain. Other studies have reported a higher pain level after radical excision surgery—48 mm; however, in the aforementioned study, pain intensity was measured while changing the patients’ bandages, which was shown to be one of the more painful experiences [23].

While assessing pain intensity with pressure algometry before and 1 day after surgery, PPT and the MTPP in different pain points decreased. A statistically significant difference (*p* < 0.05) in pain intensity levels was determined between the surgical groups 1 day after surgery when comparing PPT on the left, the MTPP on the left and center, and the IPPT at the center of the surgical wound. These variables were significantly higher in the excision group than in the “pit-picking” group. This indicates that by applying pressure directly to the wound with the pressure algometer after “pit-picking” surgery, patients tolerate pain less than in the excision group. These results may be due to the fact that during “pit-picking” surgery, the wound is sutured, and by applying pressure, tension of the tissues is created, activating the nociceptors, in contrast to the radical excision group, where the wound is left unsutured [24].

While our study aimed to determine the potential correlation between the VAS and pressure algometry to justify the use of pressure algometry for more objective postoperative pain assessment, our results show a statistically insignificant relationship between the aforementioned pain assessment methods. This raises the following question: is objective pain assessment always effective and needed? [25]. A more accurate postoperative pain assessment method, that is, pressure algometry, is still not commonly used in practice. No standardized guidelines on the use of pressure algometry have been published, although some authors provide recommendations for the standard application of this method [26]. Other authors, similar to our study, use this method to objectify the pain experience in evaluating the benefits of transcutaneous electrical nerve stimulation (TENS) in treating postoperative pain [27]. Additionally, this pain assessment method requires specialized equipment and training, which can be costly and time-consuming for healthcare providers, causing inconvenience for rapid pain assessment and interpretation of the obtained results. Considering that the difference in pain intensity measured by pressure algometry was insignificantly different from the VAS, pressure algometry seems to introduce more inconveniences than benefits.

However, this study has some limitations that must be considered when interpreting the results. The primary limitations of our study lie in its relatively small sample size, single medical center study design, and the absence of patient randomization, although the groups were homogeneous. This study included only adult patients of a specific ethnic group, excluding those diagnosed with pilonidal disease before the age of 18 years. It should be noted that pain was evaluated only on the first postoperative day, which does not allow for long-term follow-up after full recovery. Subjective evaluation of pressure algometry is possible because the bystander evaluator is involved. Furthermore, the results of pressure algometry can only be interpreted as increased or decreased pressure tolerance and do not illustrate the intensity of pain. In contrast, the visual analog scale (VAS) indicates minimal, moderate, or high pain levels. In addition, it should be taken into account that no significant relationship between the VAS and pressure algometry does not disregard the fact that pressure algometry is an objective method of pain assessment. This study may be expanded to compare pressure algometry and the visual analog scale in measuring pain intensity while the patient is in motion, a period when patients report higher pain intensity [12].

## 5. Conclusions

In the early postoperative period following pilonidal cyst surgery of varying extents, pain measured with the VAS does not differ. In contrast, the pressure algometry method showed greater pain in the minimally invasive surgery group on the first postoperative day. However, further larger studies are needed to compare these pain assessment methods in reporting pain intensity experienced during patient movement. 

## Figures and Tables

**Figure 1 medicina-60-01710-f001:**
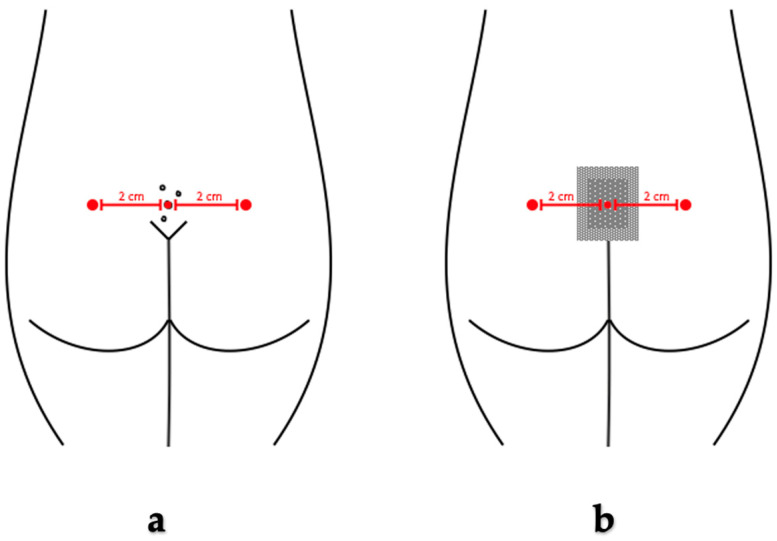
Algometry measurement points at the site of the pilonidal cyst (**a**) before surgery and (**b**) after surgery.

**Figure 2 medicina-60-01710-f002:**
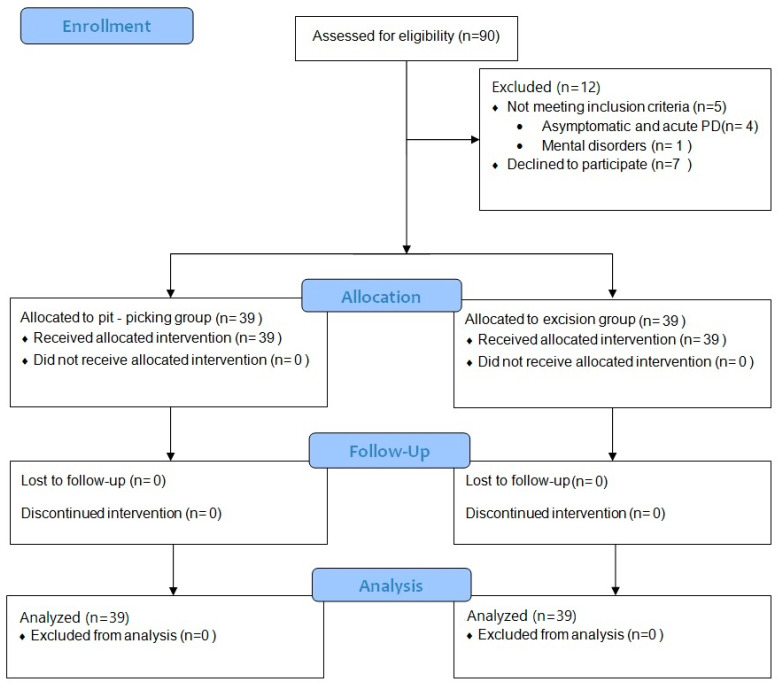
Flow diagram.

**Table 1 medicina-60-01710-t001:** Baseline characteristics.

Baseline Characteristic Item	Excision Group (N = 39)	Pit-Picking Group (N = 39)	*p* Value
Male sex, *n* (%)	33 (84.6%)	35 (89.7%)	0.46
Age *	26 (26.44–32.13)	26 (26.46–33.28)	0.83
BMI *	26.47 (25.87–29.82)	27.31 (26.19–28.71)	0.43
Smoking, *n* (%)	18 (46.2%)	23 (59.0%)	0.28
Duration of symptoms, months *	6 (9.38–25.59)	12 (11.20–24.69)	0.56
ASA class, *n* (%)	I—33 (84.6%) II—6 (15.4%)	I—36 (92.3%) II—3 (7.7%)	0.28
Primary PD, *n* (%)	29 (74.4%)	32 (82.1%)	0.41
Size of PD—cm *	5.5 (4.65–6.24)	5.2 (4.97–6.83)	0.46
Count of grown follicles *	3 (2.25–3.13)	3 (2.58–3.79)	0.29
Fistula *n* (%)	28 (71.8%)	24 (61.5%)	0.23

* Median and confidence interval. BMI—body mass index, ASA—American Society of Anesthesiologists, and PD—pilonidal disease.

**Table 2 medicina-60-01710-t002:** Results of pain intensity measurement by VAS before and after surgery.

	Excision Group (N = 39)	Pit-Picking Group (N = 39)	*p* Value
VAS before surgery *	2 (4.72–11.84)	4 (5.45–15.01)	0.29
VAS during anesthesia *	40 (37.3–48.4)	35 (27.8–40.4)	0.06
VAS I day after surgery *	13 (14.36–28.46)	15 (15.02–27.23)	0.62

* Median and confidence interval. VAS—visual analogue scale.

**Table 3 medicina-60-01710-t003:** Results of pain intensity measurement by pressure algometry before surgery.

	Excision Group (N = 39) *	Pit-Picking Group (N = 39) *	*p* Value
PPT right before surgery (kg/cm^2^)	3.17 ± 1.33	3.38 ± 1.21	0.46
MTPP right before surgery (kg/cm^2^)	4.45 ± 1.26	4.74 ± 1.30	0.32
IPPT right before surgery (kg/cm^2^)	1.26 ± 0.68	1.41 ± 0.70	0.34
PPT left before surgery (kg/cm^2^)	2.94 ± 1.22	2.94 ± 1.22	0.98
MTPP left before surgery (kg/cm^2^)	4.19 ± 1.20	4.28 ± 1.22	0.76
IPPT left before surgery (kg/cm^2^)	1.32 ± 0.78	1.43 ± 0.92	0.57
PPT center before surgery (kg/cm^2^)	2.08 ± 1.15	2.07 ± 1.02	0.96
MTPP center before surgery (kg/cm^2^)	3.01 ± 1.35	3.00 ± 1.28	0.98
IPPT center before surgery (kg/cm^2^)	0.91 ± 0.57	0.93 ± 0.57	0.86

* Mean ± SD. PPT—pressure pain tolerance, MTPP—maximum pressure pain tolerance, and IPPT—interval of pressure pain tolerance.

**Table 4 medicina-60-01710-t004:** Results of pain intensity measurement by pressure algometry after surgery.

	Excision Group (N = 39) *	Pit-Picking Group (N = 39) *	*p* Value
PPT right I day after surgery (kg/cm^2^)	1.99 ± 1.06	2.23 ± 1.02	0.31
MTPP right I day after surgery (kg/cm^2^)	3.23 ± 1.34	3.49 ± 1.23	0.39
IPPT right I day after surgery (kg/cm^2^)	1.29 ± 0.71	1.24 ± 0.62	0.91
PPT left I day after surgery (kg/cm^2^)	2.05 ± 1.46	1.42 ± 0.73	0.02
MTPP left I day after surgery (kg/cm^2^)	2.91 ± 1.33	2.32 ± 1.14	0.04
IPPT left I day after surgery (kg/cm^2^)	0.86 ± 1.31	0.90 ± 0.66	0.83
PPT center I day after surgery (kg/cm^2^)	1.50 ± 0.67	1.32 ± 0.68	0.23
MTPP center I day after surgery (kg/cm^2^)	2.51 ± 1.07	1.91 ± 0.91	0.01
IPPT center I day after surgery (kg/cm^2^)	0.98 ± 0.62	0.59 ± 0.39	0.00

* Mean ± SD. PPT—pressure pain tolerance, MTPP—maximum pressure pain tolerance, and IPPT—interval of pressure pain tolerance.

## Data Availability

The data presented in this study are available on request from the corresponding author.

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
