# Peer review of "A Comparative Analysis of Pain Assessment Methods in the Initial Postoperative Phase Following Different Pilonidal Cyst Surgeries"

_medicina, 2024, doi:10.3390/medicina60101710_

Round 1

Reviewer 1 Report

Comments and Suggestions for Authors

The authors evaluated the efficacy of pressure algometry and the visual analogue scale in measuring pain intensity in patients undergoing surgery for pilonidal cysts of varying severity and investigated possible correlations between two surgical methods (pit-picking surgery vs. radical excision) in 78 patients from a single centre. They concluded that the pain measured with the VAS in the early postoperative phase did not differ after surgery on pilonidal cysts of varying extent. In contrast, the pressure algometry method showed greater pain on the first postoperative day in the minimally invasive surgery cohort.

1. Abstract – It is not necessary to include the name of the institution in abstract. In addition, and more importantly, the results are poorly presented in the abstract. For each comparison variable, numeric/categorical results should be presented along with p-values. General statements with p-values are not acceptable. In addition, exact p-values should be reported instead of p<0.005 or p>0.005.

2. Introduction – The authors state that the specific surgical area is the perineal and sacrococcygeal region. As they are investigating painful pilonidal disease, they should not refer to the perineal region as pilonidal cysts are not localised in the perineal region.

3. The methodology is inadequately presented and lacks several important pieces of information (see comments 4 – 14). In addition, the methodology should be presented in a more reader-friendly way. I would recommend dividing the methodology into several chapters as follows: 2.1. Patients; 2.2. Ethical aspects; 2.3. Study outcomes; 2.4. Study design; 2.5. Description of the procedure (surgical and pain control); 2.6. Follow-up; 2.7. Statistical analysis.

4. The first sentence in the Methods section (lines 92 – 94) should be moved to the end of the Introduction.

5. Institutional Review Board statement – please include the date of approval next to the approval number.

6. The authors have only stated the inclusion criteria of the study. The exclusion criteria should be listed clearly after the inclusion criteria.

7. Outcomes of the treatment – the authors should clearly define the primary and secondary outcomes of the study and present them in a separate paragraph in the methodology.

8. The surgical procedure is not described once and should be presented in the methodology or appropriate references for each surgical procedure should be provided. In addition, the manuscript would benefit from intraoperative photographs of each technique, if available.

9. It is unclear what criteria were used to assign patients to each study group; as far as I know there was no randomization? This should be described in detail.

10. All variables measured should be described clearly and in detail in the methodology under the study design (e.g. what demographic or clinical data were measured for each patient, etc.).

11. Pain control is not even described in the methodology. Follow-up including pain control and discharge criteria should be described.

12. This study is flawed in its basic design. Regarding the study design, it is unclear why pain was only measured one day after surgery and not two or even more more days to allow for long-term follow-up. In this way, the significance of the study and its scientific and clinical value are questionable.

13. The statistical analysis – The IBM SPSS Statistics 29.0 should be presented correctly. Please revise the text as follows: Statistical Package for the Social Sciences (SPSS, version 29.0, IBM Corp., Armonk, NY, USA).

14. A flowchart of the study should be presented in the methodology.

15. The first sentence in the Results section (lines 157 – 159) should be deleted as the same sentence has already been mentioned in the Methods. There is no need to repeat the same data.

16. The tables should be revised for better clarity. The tables should be self-contained and easy to understand without the need to refer to the main text, which increases the clarity and accessibility of the data presented. Any abbreviation that appears in a table should be explained in a table legend, regardless of whether it was previously mentioned in the main text.

17. Spearman correlation data should be presented in a separate table.

18. Exact p-values in the text of the results section should be given instead of p<0.005 or p>0.005.

19 The single centre design should be mentioned under study limitations.

20. The manuscript would benefit from English proofreading.

21. Finally, I do not see any new findings or conclusions in this study that have not already been published in the literature. The authors should indicate how this study differs from previously published data.

Comments on the Quality of English Language

The manuscript would benefit from English proofreading.

Author Response

Dear Reviewer,

Thank you very much for your thoughtful and thorough review of our research article, "Comparative Analysis of Pain Assessment Methods in the Initial Postoperative Phase Following Different Pilonidal Disease Surgeries." We are grateful for your time and insight, which have contributed to strengthening our work.

  1. We deleted name of the institution in the abstract. However, it is impossible to provide exact p values in the abstract as we talk about generalized result and not a specific number or measurement.
  2. We agree with this comment and have made the changes in the article.
  3. We have corrected the article in response to this comment.
  4. We have corrected the article in response to this comment.
  5. We have corrected the article in response to this comment.
  6. We have corrected the article in response to this comment.
  7. This study is limited to measuring pain only during the first postoperative day, so we cannot evaluate what are the outcomes of the treatment. Especially because our study did not analyze outcomes of the treatment of pilonidal disease.
  8. The surgical procedures are standardized throughout the world and the study did not deviate from these standard procedures, because this study was not about the treatment methods of pilonidal disease. The key information about difference in pain pathophysiology after these surgeries is provided in the introduction and discussion sections.
  9. For patients’ division into two groups randomization was not used. Patients were divided into those groups based on the surgery type that was assigned to each patient during their consultation before hospitalization.
  10. We have corrected the article in response to this comment.
  11. We added information in results section, which states that “Regarding the need for analgesics, no statistically significant difference was found between the two groups of patients on the first postoperative day.”
  12. The goal of this study was to compare two pain evaluation methods, not to follow the outcomes of the pilonidal disease treatment. That is why pain was measured only during first postoperative day.
  13. We have corrected the article in response to this comment.
  14. The study was not randomized, all patient, who agreed to participate in the trial, did. So there is no purpose for a flow chart.
  15. We have corrected the article in response to this comment.
  16. We have made the changes to make tables in the article better and easier to understand.
  17. We do not see the need to provide a table about correlation data as it is presented in text that no statistically significant data was wound trying to find correlation between these methods.
  18. In result section we talk about summarized results and main findings, so we do not present exact p-values for exact measurements. Exact p-values can be found in the data tables.
  19. We have corrected the article in response to this comment.
  20. We have corrected the article in response to this comment.
  21. Thank you for your insights and thorough review of our article. In our opinion, postoperative pain is a highly relevant topic, and the objectification of pain assessment could serve as a basis for further research into postoperative pain. While our study is not very large, the results highlight an important detail – the VAS method is not entirely adequate for pain assessment, especially during physical activity. Therefore, the pressure algometer could be used in larger-scale studies on postoperative pain.

Reviewer 2 Report

Comments and Suggestions for Authors

The authors conducted a comparative study about two pain scoring methods including VAS and pressure algometry. The included 78 subjects and screened their pain scores with both of the methods before, during, and after surgery. However, the topic is interesting and the PAIN is a potential keyword both in clinical healthcare and scientific community, the paper needs a minor revision for enhancement. The comments are represented in the bellow:

Introduction:

The introduction is suitably written, but it can be improved. When we are talking about Pain, specifically in surgical situations, the Pain Killers like Morphine should be considered as alleviating agents of patients’ care. This important item is absent in this study which should be added. The authors are suggested to add a brief in the Introduction about Pain killers, Opioids, and Addictive behaviors of postsurgical managements [better in line 70].

Materials and Methods:

The inclusion criteria lack the medication consumption both by patients and the pre-surgical environment. Please add the Analgesic drug status of the patients before your measurement [lines 100-102].

Results:

In table 2, the p-value of VAS during Anesthesia is a kind of marginal non-significant and can be mentioned and then the authors can add a statement in the conclusion or limitation that with a larger sample size this item might be significant in the future studies.

Discussion:

-Adding the sample size of references in the Discussion section will make your study clearer.

-The authors talked about the lack of proper guidelines for Algometry pressure measurement but they did not discuss about their own suggestions.

-Please add comprehensive statements about the time- and cost-saving benefits of both VAS and Algometry procedure.

Author Response

Dear Reviewer,

Thank you very much for your thoughtful and thorough review of our research article, "Comparative Analysis of Pain Assessment Methods in the Initial Postoperative Phase Following Different Pilonidal Disease Surgeries." We are grateful for your time and insight, which have contributed to strengthening our work.

Introduction – in our opinion, additional brief about painkillers, opioid, etc. would be redundant information as our study did not analyze consumption of painkillers after the surgery.

Materials and Methods – thank you for your comment. All the patients in the study were classified as I-II ASA class, they don’t have any comorbidities and do not use any medications daily.

Results – in our opinion, the changes based on this comment are not needed as they do not change the essence of this article.

Discussion:

- In our opinion, adding sample size of references in the discussion section does not change the essence of this article.

- In the conclusions it is stated that further larger research is needed to evaluate possibilities of these methods. With the results of our small study, we cannot provide generalized suggestions about using pressure algometer for pain evaluation.

- Study was too little to reach a reliable conclusion about time- and cost- saving benefits of these two methods. However, the most common pros and cons are discussed in the introduction.

Reviewer 3 Report

Comments and Suggestions for Authors

After critically reviewing this Research Article titled "Comparative Analysis of Pain Assessment Methods in the Initial Postoperative Phase Following Different Pilonidal Cyst Surgeries", I found the article quite interesting, which determined my "ACCEPT" recommendation. Below you will find my detailed comments.

The authors proposed to compare two methods of assessing postoperative pain in different pilonidal cyst surgeries: VAS and algometer. The sample consisted of 78 patients divided into two groups with different degrees of surgery.

There were statistical differences only by the type of surgery, but not between the methods of evaluating postoperative pain, which proves that VAS is a non-invasive, low-cost and accessible method for all healthcare professionals.

The research lasted 3 years and included a reasonable number of patients, according to the sample calculation performed by the authors.

Although simple, the research provided the answers it set out to provide.

Reviewer 4 Report

Comments and Suggestions for Authors

Dainius, et al compared the pain intensity between patients who had undergone pilonidal cyst excision surgery compared to a pit-picking surgery using VAS and pressure algometry.   I have some suggestions for improvement:

·         The title and statement of objectives do not accurately reflect what the study evaluated.  Based on the study design and the reporting of the results, the objective of the study appears to be to compare the level of pain, both at rest (VAS) and evoked (pressure algometry), between patients undergoing excision surgery and minimally invasive pit-picking surgery.  If the authors wished to compare the utility of the two pain assessment techniques, they would have needed to evaluate which was a better indicator of some clinical outcome (e.g. analgesic use, recovery time, complications, other patient-reported outcomes, etc).  Please revise the title and text to clarify this.

·         The VAS was used to assess pain at rest, while pressure algometry is measuring evoked pain.  Thus, it is not surprising that these measures were not correlated.  Did the authors assess pain during dressing changes/sitting/using the bathroom by VAS also?  If so, was this better correlated with the pressure algometry measures?  One could hypothesize that these might have a stronger correlation because they are both assessments of evoked pain.

·         Please include a graph showing the data used for the correlation analysis.

·         Were any analgesics given after the surgery that might have impacted the pain assessments on Day 1?  If so, please provide this information in the text or table and discuss how this might have impacted the results.

·         In Tables 3 and 4, please indicate the units of the pressure algometry measurements.

·         On lines 109-110, there is a reference to an average postoperative pain score of 2.5 ± 1.7 SD.  This was reported on a 0-10 VAS, rather than the 0-100 VAS used in the current study.  Please clarify this in the text.

·         On line 74, the sentence “One specific surgical area is the perineal and sacrococcygeal region.” does not make sense in the context of the rest of the paragraph.  It would be more relevant for this sentence to describe pilonidal cyst disease.  Please revise.

Author Response

Dear Reviewer,

Thank you very much for your thoughtful and thorough review of our research article, "Comparative Analysis of Pain Assessment Methods in the Initial Postoperative Phase Following Different Pilonidal Disease Surgeries." We are grateful for your time and insight, which have contributed to strengthening our work.

  1. We want to clarify that the objective of this study was to compare two different pain assessment methods but not the pain levels. We corrected the stated goal in the article and added comment to the results section about consumption of painkiller after surgery.
  2. Thank you for the correct comment. Therefore, in the discussion section we stated that there is no correlation between VAS and pressure algometry because VAS was measured at rest while pressure algometry evaluates evoked pain. This study did not measure VAS during any physical activity so we cannot compare the results.
  3. We did not provide this graph because no statistically significant correlation was found. We stated this in the text and think that it is unnecessary to additionally analyze these results.
  4. We added information in results section, which states that “Regarding the need for analgesics, no statistically significant difference was found between the two groups of patients on the first postoperative day.”
  5. We have corrected the article in response to this comment.
  6. This comment is not accurate, as we cite literature data in the article for our sample size calculation. The pain assessment using a VAS scale of 2.5 cm or 25 mm will not affect our sample size.
  7. We have corrected the article in response to this comment.

Reviewer 5 Report

Comments and Suggestions for Authors

This study refers to a very common topic of general surgery, the surgical treatment of symptomatic pilonidal cysts. It is of interest that the authors utilize both visual analog scale and pressure algometry in pain assessment. 

Abstract 
The abstract is concise. I would propose the addition of an extra sentence in the background session, that will explain the problem of pain associated with pilonidal cyst surgery.
Introduction
- The information provided in this section is valuable for the comprehension of the manuscript.
- The objective of the study is clearly mentioned in the last paragraph. I would suggest the 5th paragraph to be presented as the first one, since the general idea of the study refers to pilonidal cysts. Theory around pain follows.
Methods
- The study design is well explained. A major drawback is the non-explication of the separation of the patients in the two groups. Was it randomized? Was it non-randomized and what were the criteria for this selection?
- The inclusion criteria and statistical analysis are correctly analyzed.
Results
- The results are presented in an extensive and explanatory way.
Discussion
- The discussion is of good quality and includes updated data. 
- The authors inform extensively the reader about the study limitations. 
Conclusion
From the presented data, the conclusion is complete and represents the work that the authors did.

Comments on the Quality of English Language

The use of English language is good and minor editing might be needed.

Author Response

Dear Reviewer,

Thank you very much for your thoughtful and thorough review of our research article, "Comparative Analysis of Pain Assessment Methods in the Initial Postoperative Phase Following Different Pilonidal Disease Surgeries." We are grateful for your time and insight, which have contributed to strengthening our work.

Abstract – as the problem of pain associated with pilonidal disease surgery is explained in the introduction, we do not see a pint to expand our abstract with this information.

Introduction – thank you for your suggestion. Though our study concentrates on the search of pain assessment methods and its ability to objectively evaluate pain. This is why we start the article speaking about pain, its evaluation and management. Pilonidal disease surgery was only an instrument for pain evaluation, that is why we talk about it at the end of the introduction.

Methods – for patients’ division into two groups randomization was not used. Patient were divided into those groups based on the surgery type that was assigned to each patient during their consultation before hospitalization.

Round 2

Reviewer 1 Report

Comments and Suggestions for Authors

The authors have not responded adequately to my comments and I see no significant improvements in this study.

I see no improvements in the abstract. The abstract is inadequately designed. The abstract should contain results, not general statements.

The authors were asked to provide outcomes of the study. They replied that this study only measured pain on the first postoperative day, so they cannot judge what the results of the treatment were. Every study has an outcome of the treatment. The primary outcome of the study is the most important parameter investigated. The secondary outcomes are all the other parameters studied.

The authors were asked to provide a description of the surgical procedure in the methodology or provide appropriate references for each surgical procedure. Their response is really premature. They replied that the surgical procedures are standardized around the world and the study did not deviate from these standard procedures because this study was not about the treatment methods of pilonidal disease. Dear authors, this should be stated in the methodology, with references supporting the description of the procedure, or the procedure should be described.

The authors were asked about randomization. They responded that there was no randomization and that patients were assigned to study groups based on the type of surgery assigned to each patient at the pre-hospital consultation. This should be clearly stated in the methodology.

The authors were asked about pain management and follow-up. Their response is inadequate and they prematurely replied that no statistically significant difference was found between the two groups of patients on the first postoperative day in terms of analgesic requirements. Is this serious? Pain management and follow-up should be described in the methodology!

This study is flawed in its basic design. Regarding the study design, it is unclear why pain was only measured one day after surgery and not two or more days to allow for long-term follow-up. The significance of the study and its scientific and clinical value are therefore questionable. The authors did not respond appropriately to this important question.

Even though there was no randomization, a flowchart of the study should be presented in the methodology.

Finally, I do not see any new findings or conclusions in this study that have not already been published in the literature

Comments on the Quality of English Language

The manuscript would benefit from English proofreading.
